# Analysis of Progressive Muscle Relaxation on Psychophysiological Variables in Basketball Athletes

**DOI:** 10.3390/ijerph192417065

**Published:** 2022-12-19

**Authors:** Marina Pavão Battaglini, Dalton Müller Pessôa Filho, Sandra Leal Calais, Maria Cristina Oliveira Santos Miyazaki, Cassiano Merussi Neiva, Mário C. Espada, Mayra Grava de Moraes, Carlos Eduardo Lopes Verardi

**Affiliations:** 1Graduate Program in Developmental Psychology and Learning, Faculty of Science, São Paulo State University, Bauru 17033-360, Brazil; 2LAPEEF, LABOREH and MEFE Laboratories, Department of Physical Education, Faculty of Science, São Paulo State University, Bauru 17033-360, Brazil; 3Graduate Program in Human Developmental and Technologies, Biosciences Institute, São Paulo State University, Rio Claro 13506-900, Brazil; 4Department of Psicology, São José do Rio Preto Medical School, São José do Rio Preto 15090-000, Brazil; 5College of Medicine, Ribeirão Preto University, Ribeirão Preto 14096-900, Brazil; 6Instituto Politécnico de Setúbal (ESE-CIEF, ESTSetúbal-CDP2T), 2914-504 Setúbal, Portugal; 7Life Quality Research Centre (CIEQV-Leiria), Rio Maior, 2040-413 Santarém, Portugal

**Keywords:** progressive muscle relaxation, anxiety, mood states, stress, athletes, heart hate, blood pressure

## Abstract

The purpose of this study was to evaluate the effects of a progressive muscle relaxation program on psychological (stress levels, anxiety, and mood states) and physiological (blood pressure and heart rate) variables in basketball athletes. Fifty-nine basketball players, aged 14 to 19 years, and members of Bauru Basket team, were recruited for this study and grouped into control group (CG, *n* = 30) and intervention group (IG, *n* = 29). The participants were evaluated, before and after the intervention, by the following means: Competitive State Anxiety Inventory-2 (CSAI-2), to measure the pre-competitive anxiety state, i.e., how anxious each athlete felt just before the competition; Brunel Mood Scale (BRUMS), an instrument for early detection of overtraining syndrome; Recovery-Stress Questionnaire for Athletes (RESTQ-Sport), to identify the extent to which each athlete was physically or mentally stressed and the athlete’s current capability for recovery; Athlete Burnout Questionnaire (ABQ), specifically developed for athletes. The IG athletes participated in 12 progressive muscle relaxation sessions, a practice for tensing or tightening a specific muscle until an ideal state of relaxation is reached. Each participant had blood pressure and heart rate measured before and after every session. The CG athletes were evaluated similarly to those in the IG but without relaxation. The results showed statistically significant differences in cognitive anxiety (*p* = 0.039) and specific stress (*p* = 0.016) between CG and IG before the intervention; in addition, a significant heart rate decrease (*p* < 0.01) between IG and CG was noted.

## 1. Introduction

Athletes and coaches perceive emotions, particularly performance anxiety, as one of the most important factors that can influence the outcome of a sports competition. Emotions and anxiety in sports are relevant topics that raise many questions and the relationship between them and sports performance is not unanimous in the literature [1]. Anxiety and stress are often associated with competitive sports and can compromise athletes’ performance because excessive levels of physiological activation and anxiety tend to cause muscle tension, fatigue, difficulty in coordination, and reduced concentration [2,3,4].

Participation in competitive sports can be stressful and influence the athlete’s mood and anxiety. This, in turn, affects enjoyment, adherence, and participation in sports. According to Akesdotter et al. (2020) [5], 19.5% of elite athletes experience anxiety and/or depression symptoms, and eating disorders and self-reported stress, such as burnout, are also common. Competitions can also increase anxiety levels and affect the performance of athletes from different sports and categories. For example, Hoover et al. (2017) [6] observed that competitions significantly affected anxiety and performance measures in high school basketball players.

Jacobson’s progressive muscle relaxation is an important ally for various populations among the various intervention techniques used to reduce anxiety, tension, and stress. This technique is recommended in daily tensions related to various factors, such as work, interpersonal relationships, and daily routines. Several studies [7,8,9,10,11,12] were conducted to verify the relaxation effects under a diversity of emotional and physiological factors. For example, beneficial effects were noted in people with stress, anxiety, depression, sleeping problems, tension headaches, excessive worry, fear, gastrointestinal problems, high blood pressure, pain, and chronic diseases. Although the Jacobson relaxation constitutes an effective intervention in reducing emotional distress [13], its effects on athletes’ psychological skills are not yet consistent in the literature [14].

Thus, it is necessary to identify appropriate strategies and intervention programs during a competitive season to contribute to reducing the levels of anxiety and stress and improving sports performance [15,16,17]. This study aimed to analyze the effects of Jacobson’s Progressive Muscle Relaxation on psychophysiological variables in male basketball athletes. According to the propositions in the literature, the present study tested the first hypothesis, that athletes participating in a progressive muscle relaxation program have lower levels of stress and state anxiety and higher levels of perceived recovery. In the second hypothesis tested, it was considered whether progressive muscle relaxation after a training session was associated with reduced signs indicative of cardiovascular stress, such as lower heart rate (HR) and lower blood pressure (BP).

## 2. Materials and Methods

Fifty-nine male basketball athletes, aged between 14 and 19 years, from the Under-15, Under-16, and Under-19 basketball teams of Bauru Basket participated in this study. Of the participants, 29 athletes (15.97 ± 1.88 years) constituted the intervention group (IG), that participated in the progressive muscle relaxation program, and 30 athletes (14.33 ± 0.48 years) formed the control group (CG). The sample was chosen for convenience, according to the teams’ availability. The inclusion criteria were having participation equal to, or greater than, 75% of the number of sessions planned in the intervention program, all athletes undergoing the same training program, and not having a history of progressive muscle relaxation training. 

The IG and CG athletes filled out a demographic evaluation form before the intervention program, consisting of a questionnaire with open and closed questions about the following variables: age, time as a federated athlete, competitive participation time, and training characteristics. The athletes in both groups were evaluated before, and at the end of, the 12 sessions using the following instruments:

Competitive State Anxiety Inventory-2-CSAI-II [18,19]. This aims to measure pre-competitive state anxiety, that is, how anxious an athlete feels at a given moment in time. This instrument consists of 27 questions, divided into three subscales: cognitive anxiety, somatic anxiety, and self-confidence, which are assessed on a Likert scale, in which the subject chooses from four grades, corresponding to the subject’s momentary emotional state, namely: 1-nothing, 2-somewhat, 3-badly, and 4-completely, according to each question. The subscale scores were obtained by summing the answers, ranging from 9 to 36. For interpretation, the data on cognitive, somatic, and self-confidence anxieties were categorized as low (9 to 18 points), medium (19 to 27 points), and high (28 to 36 points).

Stress and Recovery Questionnaire for Athletes-RESTQ 76 Sport. This was developed to assess an athlete’s current stress and recovery profile [20], and was validated for the Portuguese language by Costa; Samulski (2005) [21]. It consists of a series of statements that indicate athletes’ mental and emotional states and physical well-being. Responses were given on a Likert-type scale: 0. Never; 1. Very infrequently; 2. A few times; 3. Half the time; 4. Many times; 5. Very often; 6. Always. This questionnaire was developed to measure the frequency of the current state of stress in conjunction with the frequency of associated recovery activities. To this end, it assesses potentially stressful events and recovery phases and their subjective consequences over the past three days and nights [20]. The internal consistency of the instrument was between α = 0.58 and α = 0.85.

According to the scale ranging from 0 to 6, values above 4 were considered “high,” below 2 were “low,” and between 2.01 and 3.99 were “moderate,” either for specific stress, general stress, or recovery. The scale values were calculated by the mean values of the respective items. High scores on the scales associated with stress activities reflected intense subjective stress, while high scores on the scales associated with recovery reflected many recovery activities. In general, low scores in stress-related areas and high scores in recovery-related areas are considered positive, and vice versa [20]. 

### 2.1. Procedures

Initially, the research project was forwarded to the Research Ethics Committee and approved according to the process (CAAE: 68506417.1.0000.5398; Opinion number: 2.224.815). Prior to data collection, permission to develop the research was requested from the professionals in charge of the participating team, and the objective and methodology to be used during the study were presented. The participating athletes were informed about the conditions for entering the research and received the standardized instructions and information.

The Informed Consent Form (ICF) and the Free and Informed Assent (TALE) for the athletes were previously sent to the parents of the athletes under 18 years of age who agreed to participate in the research. The conditions for participating in the research were the return of the Informed Consent Form, duly signed by parents or guardians, and the Free and Informed Assent signed by the athletes. The athletes over 18 years old agreed to participate in the research by signing the Free and Informed Consent Form. After training, the athletes answered the questionnaires in the university’s classroom, where the training sessions took place.

### 2.2. Intervention Program

The athletes from the intervention group (IG) participated in 12 sessions with Edmund Jacobson’s (1938) Progressive Relaxation technique, which consisted of retightening (making tense) and relaxing specific muscles until they were all relaxed [22]. To obtain the desired therapeutic relaxation, studies indicate that approximately 12 sessions are needed with the training of the technique [23,24,25]. The technique was applied and supervised collectively (with a maximum of 10 athletes per session) according to Sandor’s (1982) [22] adapted script, once or twice a week after training and lasting between 30 and 40 min per session, for the athletes in the intervention group, for all sessions. The relaxation sessions took place in a room prepared and appropriate for the procedure, using mats.

The intervention group (IG) and the control group (CG) completed all the evaluation instruments, Competitive State Anxiety Inventory-2 (CSAI-II) and Stress and Recovery Questionnaire for Athletes (RESTQ 76 Sport) at two different times, on the previous day at the beginning of the intervention, and on the day after the end of the 12 sessions of progressive muscle relaxation. BRUMS and RESTQ 76 Sport, take about 15 and 20 min to be completed, depending on the familiarity with the questionnaire. Blood pressure (BP) and heart rate (HR) were recorded by using an automated digital (Omron HEM-7122, JAP). Blood pressure (BP) and heart rate (HR) were measured in a standardized way; three measurements were recorded on the left arm, the cuff was positioned at heart level, the palm of the hand facing up, the back and the forearm were supported, and the legs and feet supported on the floor. During the 12 progressive muscle relaxation sessions, the control group (CG) remained on the court, accompanied by the coach and did not perform any activity. After the end of the intervention period, the CG members were invited to participate in the progressive muscle relaxation program, according to the rules of ethical commitment.

### 2.3. Data Analysis

To evidence the internal consistency for each instrument used, Cronbach’s Alpha Coefficient (α) was calculated. Reliability was assessed, as proposed by George and Mallery (2003): excellent (>0.90); good (>0.80); acceptable (>0.70); questionable (>0.60); poor (>0.50); unacceptable (<0.50). For data analysis, the following descriptive statistics were calculated: mean (x); standard deviation (SD); median (Md); and quartiles (Q1-Q3). To verify data normality, an exploratory analysis was performed using the Kolmogorov–Smirnov test, with statistical significance set at *p* < 0.05. As the data did not meet the normality propositions (*p* < 0.05), the Mann–Whitney non-parametric test was used to compare the intervention group (IG) with the control group (CG) pre- and post-intervention. The data were analyzed using the software IBM^®^ SPSS^®^ Statistics Version 25, and a *p* < 0.05 significance level was adopted in all statistical tests.

## 3. Results

Initially, Cronbach’s alpha coefficient was adopted, aiming to estimate the reliability and internal consistency of the instruments used in this study. A total result of Alpha was observed for CSAI-2 (0.60) and RESTQ-76 (0.79); therefore, the results were considered acceptable. 

As shown in Table 1, the IG athletes had an average of four years as federated athletes, and the CG athletes had only two years. Both the IG and the CG maintained their training routine during the study, so the IG kept training, on average, six times a week and three hours a day, and the CG kept training, on average, five times a week and two hours a day. The IG athletes participated in six regional competitions, and the CG athletes participated in four regional competitions, on average.

An analysis performed with the Mann–Whitney test (Table 1) indicated a statistically significant difference (*p* < 0.05) between the athlete groups (intervention and control). The members of the intervention group (IG) had been federated athletes in the Basketball Federation of São Paulo for a long time. It can be noticed that the IG athletes presented higher frequencies and hours of training per week, compared to the control group (CG).

Table 2 shows the behavior of data related to state anxiety, stress, and recovery. Given the proposed classification, IG and CG athletes reported pre- and post-intervention mean and median scores of cognitive and somatic state anxieties classified as low. Self-confidence followed the same trend for both groups, with mean and median scores classified as medium pre- and post-intervention periods. The mean and median values presented (Table 2) by the Stress and Recovery Questionnaire for Athletes (RESTQ-Sport) pointed to the factors of general stress, specific stress, and global stress, classified as moderate for the intervention group and low for the control group. In general, for both groups, it was noted that the data for the Recovery factors (General Recovery, Recovery Areas, and Global Recovery) were classified between high and moderate pre- and post-intervention periods.

The analysis performed with the Mann–Whitney test (Table 3) showed a statistically significant difference in the cognitive anxiety variable (U = 299.000; *p* = 0.038) pre- and post-intervention, comparing IG and CG. However, no statistically significant differences were found in the somatic anxiety and self-confidence domains.

According to Table 3, the Mann–Whitney Test showed that the progressive muscle relaxation technique affected specific stress (U = 275.500; *p* = 0.015). However, there was no statistically significant difference between the groups of athletes for the subscales General Stress (U = 426.500; *p* = 0.897), Global Stress (U = 353.000; *p* = 0.214); General Recovery (U = 397.000; *p* = 0.564); Recovery Areas (U = 343.000; *p* = 0.163); and Global Recovery (U = 349.000; *p* = 0.192).

Table 4 provides data on heart rate (HR) behavior, pre- and post-intervention, comparing the differences in the mean HR of the two groups. The Mann–Whitney test showed a statistically significant difference (*p* < 0.05) in six of the 12 sessions. The intervention group had greater differences compared to the control group. This difference was concentrated between the 5th and 10th sessions, suggesting that the effects of progressive muscle relaxation were more pronounced in this time interval. 

The same behavior did not occur for blood pressure (Table 5). Only three sessions showed statistically significant differences (*p* < 0.05) between the groups. The differences were located in the seventh session for systolic blood pressure (U = 234.000; *p* = 0.038), and in session 2 (U = 188.500; *p* = 0.000) and session 10 (U = 67.000; *p* = 0.005) for diastolic blood pressure measurements. 

## 4. Discussion

The present study aimed to verify the effects of Jacobson’s progressive muscle relaxation on psychological (anxiety, mood states, and stress) and physiological (blood pressure and heart rate) variables in male basketball athletes. Few studies have examined the influence of relaxation techniques in sports environments, and several studies in other areas have emphasized the positive effects of relaxation on psychophysiological variables, justifying the present study [26].

The results obtained showed significant improvement in cognitive state anxiety after the intervention, but somatic state anxiety and self-confidence did not exhibit the same effect. These results partly corroborated studies [27,28,29] that indicated that relaxation was an effective technique for reducing anxiety. Bargherpour et al. (2012) [27] revealed that progressive muscle relaxation reduced somatic anxiety and cognitive anxiety and increased self-confidence in taekwondo players, as did the visualization technique. Other techniques, such as autogenic training, guided visualization [17], and mindfulness [30], appeared to reduce cognitive anxiety and increase self-confidence. 

This result was positive, as low levels of cognitive state anxiety are associated with better coping in competitive situations and emotion regulation [31]. Thus, muscle relaxation may have led to better coping in competitive situations and emotion regulation, decreasing cognitive anxiety. As Kudlackova, Eccles, and Dieffenbach (2013) [32] pointed out, progressive relaxation is often chosen by athletes to deal with competitive anxiety. However, other relaxation practices, such as meditation and visualization, are more commonly used in coping with everyday anxiety.

Athletes showed low levels of cognitive anxiety and somatic anxiety. Although this was not a consistent finding in the literature, some studies [33,34] indicated that athletes do not have high anxiety scores. It is noteworthy that data collection in this study was performed after training sessions, which may have contributed to the low levels obtained. High anxiety levels could have been found in the questionnaires answered before competitions, as studies [35,36] demonstrated higher anxiety levels before competitions than during training sessions.

Some factors may be related to the low levels of anxiety found, such as male sex, satisfaction in the sports career, and team sports. Rice et al. (2019) [37] identified younger age, recent experience of adverse life events, and dissatisfaction in sports career as factors associated with anxiety symptoms in female athletes. Furthermore, team sports are less vulnerable to anxiety than individual sports [34].

The athletes in the intervention group (IG) showed higher cognitive anxiety levels than the athletes in the control group (CG). Considering that the IG athletes had more time as federated athletes and participated in more competitions than the CG, these results were contrary to research [37,38] that indicated that athletes with more competitive experience presented lower cognitive and somatic anxiety scores, compared to younger athletes with little competitive experience. On the other hand, other studies [34,39] found no statistically significant differences regarding competitive anxiety between younger, less experienced, athletes in a sport and more experienced athletes. The higher anxiety scores found in IG compared to CG in our study might be associated with the fact that IG contained athletes who competed in the adult professional category and were subjected to more competitive games [37].

Concerning stress and recovery levels, the data agreed with Codonhato et al. (2018) [40], who found an adequate recovery profile significantly higher than stress, which is considered ideal for performance. At average stress levels, the athlete can achieve an optimal level of performance through adequate recovery. However, when stress levels are higher, athletes may become unable to meet recovery demands if they do not engage in additional recovery activities [41]. 

A statistically significant difference was observed between IG and CG only for the Specific Stress domain after the intervention, indicating that the relaxation technique may contribute to reducing this variable in athletes. It is worth mentioning that this domain is related to what is perceived specifically in situations related to sports activity: disruption in the interval (recovery deficit, interrupted recovery, and situational aspects related to the rest period), emotional exhaustion (athletes who feel saturated and psychically exhausted with the sports activity), and injuries (acute injuries and vulnerability to injuries). Thus, the decrease in Specific Stress levels after the relaxation sessions indicated that the technique could reduce emotional exhaustion and improve athletic recovery.

Several studies reinforce the importance of psychological monitoring in athletes associated with traditional physiological markers, such as cortisol, blood pressure, and heart rate [36,42,43,44]. Morales et al. (2013) [44] investigated stressful situations before judo competitions among international and national athletes and found that heart rate analysis was sensitive to changes in pre-competitive anxiety. Mateo et al. (2012) [43] confirmed that heart rate analysis provided a complementary tool for assessing competitive pressure in cycling training programs. Fortes et al. (2017) [42] suggested that athletes with a high magnitude of cognitive anxiety and/or somatic anxiety demonstrated a high degree of autonomic nervous system disturbance.

Regarding the physiological effects after progressive relaxation, Bara Filho et al. (2002) [45] stated that the technique reduced blood cortisol levels, demonstrated an intense psychophysiological relationship among the human body processes, and indicated the need to use strategies to control sports training to avoid excessive stress. For Toledo and Filho (2007) [46], Jacobson’s progressive relaxation technique could be used as an auxiliary method in the athlete’s psychophysiological recovery due to the more efficient removal of blood lactate.

Maimunha and Hashim (2016) [24] revealed results which indicated a significant reduction in heart rate and choice reaction time after relaxation with seven or 16 muscle groups. In our study, after the intervention, the experimental groups exhibited significantly lower heart rates than the control group. The reduced heart rates in both training groups suggested that progressive muscle relaxation could balance sympathetic and parasympathetic activation responses during intense activity. Since heart rate indicates cardiovascular tension, the lower heart rate found suggested the potential usefulness of progressive relaxation in delaying the onset of fatigue during sports.

In this study, only three sessions had significant differences between IG and CG for blood pressure (BP) pre- and post-intervention, and there was no significant effect when comparing the mean between the 12 sessions. The differences were not concentrated in any period, suggesting that muscle relaxation did not affect blood pressure.

For heart rate (HR), there was a statistically significant difference pre- and post-intervention when comparing the HR means of the two groups. In six out of 12 sessions, the IG had greater differences than the control group, indicating a decrease in HR. These data corroborated Rissardi and Godoy (2007) [47], who found no significant difference between the initial and final values of systolic and diastolic blood pressure over ten sessions of progressive muscle relaxation with leprosy patients; however, they did observe a significant difference in HR and respiratory rate (RR). According to the authors, since there was a drop in HR due to relaxation, the ejection volume must have increased and decreased, or the peripheral arteriolar resistance remained stable. The values must have counterbalanced each other, not allowing a significant pressure drop. Another explanation for the non-modification of BP is based on the American Society of Hypertension concept. The vast majority of patients studied were normal or borderline and, thus, did not show post-relaxation modification, as occurs with hypertensive patients [47]. In the study by Astuti, Rekawati, and Wati (2019) [48], after 11 therapy sessions (progressive muscle relaxation and music therapy) over six days, the results showed a significant decrease in systolic blood pressure but not in diastolic blood pressure.

In this study, the difference between the HR indices was concentrated between the 5th and 10th sessions, suggesting that the intervention effects were more pronounced in this time interval. In Mellado’s (2015) [49] study, from the seventh session on, all participants found the activity to be easy to perform, and this data was possibly related to learning both the exercise and the ability to relax. The largest number of participants also evaluated themselves as feeling good during the sessions, and unanimity occurred as of the sixth session.

As for practical implications, the data from this study suggested that progressive muscle relaxation may be an effective technique to reduce heart rate, cognitive anxiety, and sport-related stress in athletes. This is important because controlling stress and anxiety, having self-confidence, and having relaxation skills are considered indispensable for improving performance during training and competition [28,50,51]. However, no positive effects were observed on somatic anxiety, mood, and burnout; thus, not confirming the initial hypotheses.

Despite the contributions, some limitations should be highlighted. First, the study involved a relatively small sample size. Second, significant differences in age and length of sports experience were found between the intervention and control groups. Therefore, the demographic differences found may have been random. Thus, more studies are needed to better understand the impact of age and time of sports experience on anxiety and stress. Future studies with a randomized clinical trial and larger numbers of participants from various sports that can explore the effects of progressive relaxation after games, and at different stages of the competitive season, are recommended. 

## 5. Conclusions

Jacobson’s progressive relaxation proved to be an effective technique to reduce cognitive anxiety and stress related to sports, which can contribute to the resolution of conflicts existing in the sports context, such as competitions, trips, and overtraining, among others. Furthermore, relaxation effectively reduces heart rate, helping the athlete’s recovery.

Studies with larger numbers of participants from various sport modalities that can explore the effects of progressive relaxation after games, and at different stages of the competitive season, are recommended.

## Figures and Tables

**Table 1 ijerph-19-17065-t001:** Median, quartile, and Mann–Whitney test values referring to basketball athletes (*n* = 59) grouped according to the intervention group (*n* = 29) and control group (*n* = 30).

Variables	Intervention Group		Control Group		
	Med	Q1–Q3	Med	Q1–Q3	*p*
Age	15.00	15.00–18.00	14.00	14.0–15.0	<0.001
Time as a federated athlete	4.00	3.00–5.00	2.00	1.00–3.00	<0.001
Days of training in the week	6.00	6.00–6.00	5.00	5.00–5.00	<0.001
Number of trainings a day	2.00	2.00–2.00	1.00	1.00–1.00	<0.001
Hours training per day	3.00	2.00–2.00	2.00	2.00–2.00	<0.001

Statistically significant difference (*p* < 0.05).

**Table 2 ijerph-19-17065-t002:** Mean, standard deviation, median, and quartiles (Q1–Q3) of the three subscales of state anxiety (CSAI-2) and the six subscales of stress and recovery (RESTQ-Sport) for athletes in the intervention and control groups.

		Intervention Group		Control Group	
Variable		$\bar{x}\pm s$	Md (Q1–Q3)	$\bar{x}\pm s$	Md (Q1–Q3)
ACog.	Pre	16.59 (4.31)	16.00 (13.00–19.50)	15.53 (3.95)	15.00 (12.75–17.25)
	Post	16.69 (6.15)	17.00 (11.50–20.00)	17.13 (3.73)	17.0 (14.75–19.00)
ASom.	Pre	16.62 (3.59)	16.00 (14.00–18.50)	16.07 (2.88)	16.0 (14.75–18.00)
	Post	17.17 (4.32)	17.00 (14.00–20.00)	16.30 (3.29)	15.50 (14.00–18.25)
SC	Pre	24.90 (7.42)	27.00 (22.50–30.00)	27.07 (5.27)	27.00 (23.75–32.00)
	Post	25.66 (5.83)	26.00 (20.50–31.00)	26.70 (4.04)	26.50 (24.00–30.00)
Gen.S.	Pre	2.14 (0.84)	2.04 (1.66–2.50)	1.78 (1.02)	1.66 (0.78–2.58)
	Post	2.15 (1.18)	2.07 (1.20–2.82)	1.72 (0.92)	1.66 (1.00–2.05)
SS	Pre	2.41 (1.00)	2.67 (1.87–3.25)	1.82 (1.07)	2.04 (0.79–2.46)
	Post	2.00 (1.16)	2.00 (1.21–2.75)	1.88 (0.97)	1.83 (1.18–2.50)
GS	Pre	3.90 (8.52)	2.33 (1.87–2.93)	1.81 (0.96)	1.77 (0.99–2.59)
	Post	2.08 (1.14)	2.08 (1.18–2.86)	1.80 (0.88)	1.77 (0.92–2.30)
Gen.R.	Pre	3.74 (0.59)	3.80 (3.22–4.20)	3.95 (0.76)	3.85 (3.53–4.38)
	Post	3.62 (0.83)	3.70 (2.87–4.32)	3.71 (0.90)	3.95 (3.23–4.35)
RA	Pre	4.06 (0.82)	4.06 (3.34–4.56)	4.42 (0.87)	4.23 (3.75–5.26)
	Post	3.99 (0.93)	4.13 (3.06–4.81)	4.07 (0.93)	4.13 (3.47–4.78)
GR	Pre	3.89 (0.64)	3.96 (3.41–4.22)	4.19 (0.79)	4.05 (3.72–4.78)
	Post	3.80 (0.85)	3.85 (2.90–4.58)	2.86 (0.81)	4.04 (3.35–4.43)

Note: Cognitive Anxiety (ACog); Somatic Anxiety (ASom); Self-Confidence (SC); General Stress (Gen.S.); Specific Stress (SS); Global Stress (GS); General Recovery (Gen.R.); Recovery Areas (RA); Global Recovery (GR).

**Table 3 ijerph-19-17065-t003:** Mean pre- and post-intervention differences of the three subscales of state anxiety (CSAI-2) and the six subscales of stress and recovery (RESTQ-Sport) for athletes in the intervention and control groups, and *p*-value resulting from the comparison performed with the Mann–Whitney Test.

	Intervention Group	Control Group			
Variable	Med	Med	U	Z	*p*
ACog.	1.000	−1.000	299.000	−2.073	0.038 *
ASom.	0.000	0.000	396.500	−0.592	0.554
SC	0.000	0.000	419.500	−0.236	0.814
Gen.S.	0.070	−0.160	426.500	−0.129	0.897
SS	0.450	−0.125	275.500	−2.421	0.015 *
GS	0.380	−0.120	353.000	−1.244	0.214
Gen.R.	0.100	0.200	397.000	−0.577	0.564
RA	0.060	0.310	343.000	−1.395	0.163
GR	0.090	0.305	349.000	−1.304	0.192

* Statistically significant difference (*p* < 0.05). Note: Cognitive Anxiety (ACog); Somatic Anxiety (ASom); Self-Confidence (SC); General Stress (Gen.S.); Specific Stress (SS); Global Stress (GS); General Recovery (Gen.R.); Recovery Areas (RA); Global Recovery (GR).

**Table 4 ijerph-19-17065-t004:** Mean pre- and post-intervention differences in heart rate behavior during 12 progressive muscle relaxation sessions for athletes in the intervention and control groups, and *p*-value resulting from the comparison using the Mann–Whitney test.

	Intervention Group	Control Group			
	Med	Med	U	Z	*p*
Session 1	16.00	7.00	261.500	−2.469	0.014 *
Session 2	9.00	6.50	344.000	−1.184	0.236
Session 3	12.00	6.50	303.500	−1.624	0.104
Session 4	11.00	5.50	310.500	−1.512	0.131
Session 5	12.00	3.00	199.000	−3.299	0.001 *
Session 6	8.50	8.50	402.500	−0.273	0.785
Session 7	12.00	4.00	177.500	−2.277	0.023 *
Session 8	8.50	4.00	144.000	−2.818	0.005 *
Session 9	11.00	7.00	155.000	−2.331	0.020 *
Session 10	10.00	4.00	92.000	−2.105	0.035 *
Session 11	9.00	5.00	55.500	−1.412	0.158
Session 12	13.00	5.00	8.000	−0.816	0.414

* Statistically significant difference (*p* < 0.05).

**Table 5 ijerph-19-17065-t005:** Mean pre- and post-intervention differences in systolic blood pressure behavior over 12 progressive muscle relaxation sessions for the intervention and control group athletes and *p*-value resulting from the comparison with the Mann–Whitney test.

	Intervention Group	Control Group			
	Med	Med	U	Z	*p*
Session 1	4.50	0.00	408.500	−0.179	0.858
Session 2	3.50	7.00	298.000	−1.900	0.057
Session 3	6.00	3.00	299.000	−1.696	0.090
Session 4	8.00	4.00	339.500	−1.049	0.294
Session 5	3.00	0.50	344.500	−0.968	0.333
Session 6	4.00	8.50	354.500	−1.020	0.308
Session 7	4.00	0.00	234.000	−2.070	0.038 *
Session 8	5.00	3.00	274.000	−0.279	0.780
Session 9	5.00	−2.00	264.500	−0.118	0.906
Session 10	9.00	−1.00	85.500	−1.909	0.056
Session 11	10.50	−5.00	53.000	−1.728	0.084
Session 12	8.00	−5.00	2.500	−1.944	0.052

* Statistically significant difference (*p* < 0.05). Source: prepared by the author.

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
