# Peer review of "Analysis of Progressive Muscle Relaxation on Psychophysiological Variables in Basketball Athletes"

_ijerph, 2022, doi:10.3390/ijerph192417065_

Round 1

Reviewer 1 Report

The manuscript reports upon an intervention study using progressive muscle relaxation amongst a cohort of athletes in relation to psychological and psychophysiological variables. Whilst this is conceptually worthwhile the study appears to suffer from several significant methodological shortcomings, in particular:

·         There is a statistically significant difference between the control the intervention groups on both demographic and sport related variables. This is problematic in terms of elucidating the potential efficacy of the intervention. Why were the groups not matched?

·         Blood pressure and heart rate are somewhat limited as psychophysiological measures. Heart rate variability would provide a more meaningful insight into the phenomenon of interest.

·         The rationale for the various psychometric measures is not especially clear at present. For example, pre-competition anxiety cannot automatically be assumed to be debilitating.

Additionally, there is a lack of detail in reporting the methods employed throughout the study and I have the following questions:

·         Why was the dose of the intervention not controlled? The manuscript states that the IG underwent PMR either once or twice per week.

·         It isn’t clear whether the psychometric instruments were completed pre-post each session or just pre-post the 12 week intervention?

·         How were blood pressure and heart rate measured? What equipment was used and how?

·         What did the control group do for 30 minutes and why?

In terms of the results presented:

·         How were Cronbach’s alpha values calculated? Why are values not presented for each of the subscales within each instrument?

·         The statistical approach used doesn’t appear to make use of all of the available data.

Author Response

We hereby specify the proposed changes in accordance with the requests, referring to the work entitled “Analysis of Progressive Muscle Relaxation on Psychophysiological Variables in Basketball Athletes”, which are presented in the items below:1) According to the reviewers' observations, the Method section was expanded with information about the application of the intervention program (all alterations are highlighted in red).2) A sentence was added to the results, explaining that the two groups of athletes maintained their training routine, during the research (all alterations are highlighted in red).3) Regarding the statistically significant difference between the control and intervention groups in both demographic and sport-related variables, it is worth explaining: because it was a sample chosen by convenience, and despite the study authors had presented about the importance of making a combination between the categories, the professionals responsible for the team gave permission to carried out the research, with the condition: that the intervention group (GI), should be formed by players from Under-16 and Under-19 categories; while the control group (CG) by the players from Under-15 category. Therefore, due to the discrepancy between the ages of GI and CG groups, the statistically significant differences between the demographic variables were by chance.4) Regarding the limitations of blood pressure and heart rate as psychophysiological measures, we agree that heart rate variability would provide a broader view of this phenomenon. However, the research did not have any financial support, therefore, due to lack of resources, we tried to employ a reliable instrument, easy to apply, with low cost and widely employed in other studies.This is the information we had to present. We would like to thank you for the valuable contribution and hope to have met all the reformulations pointed out by the reviewers. We remain at your disposal for further alterations, if necessary.Yours sincerely,The authors

Reviewer 2 Report

The main objective of this study was to evaluate the effects of a progressive muscle relaxation program on psychological and physiological variables in basketball athletes.

The work is well organized and provides all the sections required by the journal, however, some assessments are made for a improvement of the document:

1. Regarding formal aspects, review the entire document regarding typographical errors, mainly at the end of the line (eg, line 26, 126, 137, 152, 160, etc.)

2. The initial sample of 59 participants, divided into a control group (CG) and an intervention group (IG), is finally established at CG=24 and IG=25, which is sufficient; the way of selecting the sample is for convenience, and when normality was violated, non-parametric tests were applied. Table 1 reports n=59; it is recommended to report the n in the following tables of results, since 10 athletes were excluded from the study.

3. 12 intervention sessions were applied, with pre- and post-test measurements for both groups. It is necessary to report how the blood pressure and heart rate recording protocol was carried out, as well as the procedure and material used for said recording (sphygmomanometer, task-force R.R, heart rate monitor R-R, etc.).

4. The statistical analysis is pertinent and robust, guaranteeing the correct obtaining of the results.

5. The results confirm some of the hypotheses, and reject another, allowing the formulation of coherent conclusions. In line 219 reference is made to tables 5 and 6; however, Table 6 is missing. It is necessary to report the HR data. It is necessary to review the tables provided (and the text where they refer), and that the titles and variables are correct.

6. Lines 212 and 213 explain stronger relationships between session 5 and 10 with respect to HR; it seems that a table is missing and the results of table 4 are not correct. The Results section needs to be reviewed and corrected.

6. Limitations and practical applications are provided.

Author Response

(The authors gave the same response as above.)

Reviewer 3 Report

The study examines the effects of the progressive muscle relaxation intervention on psychological and physiological responses in basketball athletes. The authors found some differences in pre- and post-intervention responses between the intervention and control groups. Overall, I think the manuscript is very well-written and the issue is very interesting. However, there are some issues needed to be addressed.

My first concern is regarding the main effects, that is, the pre- and post-intervention differences between the two groups in cognitive anxiety and specific stress scores (Table 3). The raw results in Table 2 show almost identical values between pre- and post-intervention in cognitive anxiety scores in the intervention group, suggesting that the significant effects observed in Table 3 are mediated mostly by the differences between pre- and post-intervention in the "control" group. The main results are thus hard to convince the readers that the intervention did do something to change cognitive anxiety. I also cannot understand how to calculate Table 3 values from Table 2. If I understand correctly, Table 3 is pre minus post values, then why ACog is 1.000 (16.59 – 16.69 = -0.1) and -1.000 (15.53. – 17.73 = -2.2), and the same for other variables.

The authors reported that some demographical variables are significantly different between the intervention and control groups, which, as mentioned by the authors, is the limitation of the study because any effects observed could be simply due to these demographical differences between the two groups. To rule out these possibilities, is it possible to try something like linear-mixed models, such that the authors can add these demographical variables into account while examining intervention effects?

I think it is good to explicitly state that the two groups of athletes keep their training routine (Number of training a day & Hours of training per day) over the period of the study (perhaps around 4-6 weeks).

Author Response

(The authors gave the same response as above.)

Round 2

Reviewer 3 Report

Regarding my concern raised previously, the authors responded “In all these cases, data analyzes were performed regarding the difference in the values of all questionnaires, their subscales and their total values between the before and after analyses.”, which is understandable. But, I still cannot understand how could the direct difference values in Table 2 do not match the values in Table 3, as the values in Table 2 are from “the summation of the values of all questionnaires (e.g., sub-tests)”. Given that, I would suggest the authors to more explicitly articulate their data analyses regarding the calculation of Table 2 and 3 in detail.

The authors acknowledge the limitation of significant demographical differences between the intervention and control groups “Second, significant differences in age and length of sports experience were found between the intervention and control groups. (line 353-354)”. I think this issue is important as any observed intervention differences could be simply due to these demographical differences. I would thus suggest the authors add a few sentences that could provide a solution for this issue in future studies.  

Author Response

Dear Reviewer 3,

Responding to your first question, the intention of the authors in Table 2 was to present the descriptive statistics values ​​(mean, standard deviation, median and quartiles) referring to the three subscales of the CSAI-2 scale, as well as the six subscales from RESTQ-Sport. This presentation, made separately from inferential statistics, was intended to facilitate the understanding of its components. In Table 3, the interest was to show whether the mean difference is equal to a particular value (usually zero) or not. For this, the values ​​of the samples (intervention and control groups) pre and post intervention were calculated by the difference (di) between each pair of values ​​(of the three CSAI-2 subscales and the six RESTQ-Sport subscales, where: di = Xpre - Xpost Therefore, the values ​​presented in Tables 2 and 3 do not correspond, for this reason.

Notes made on lines 353-354 were answered in the last version of the manuscript attached to the platform.

Thus, we conclude that we accept and respond to all suggestions and criticisms regarding the manuscript.